# The Complete Chloroplast Genomes of *Gynostemma* Reveal the Phylogenetic Relationships of Species within the Genus

**DOI:** 10.3390/genes14040929

**Published:** 2023-04-17

**Authors:** Jiaxia Gan, Ying Li, Deying Tang, Baolin Guo, Doudou Li, Feng Cao, Chao Sun, Liying Yu, Zhuyun Yan

**Affiliations:** 1State Key Laboratory of Characteristic Chinese Medicine Resources in Southwest China, Chengdu University of Traditional Chinese Medicine, Chengdu 611137, China; ganjiaxia@outlook.com; 2Institute of Medicinal Plant Development, Peking Union Medical College and Chinese Academy of Medical Sciences, Key Laboratory of Ministry of Education, Beijing 100193, China; liying@implad.ac.cn (Y.L.); guobaolin010@163.com (B.G.); missbean92@163.com (D.L.); caofeng010224@163.com (F.C.); csun@implad.ac.cn (C.S.); 3Yunnan Branch of Institute of Medicinal Plant Development, Chinese Academy of Medical Sciences, Jinghong 666100, China; tangdeying2023@163.com; 4Guangxi Botanical Garden of Medicinal Plant, Guangxi TCM Resources General Survey and Data Collection Key Laboratory, Nanning 530023, China

**Keywords:** *Gynostemma*, chloroplast genome, taxonomic, phylogeny

## Abstract

*Gynostemma* is an important medicinal and food plant of the Cucurbitaceae family. The phylogenetic position of the genus *Gynostemma* in the Cucurbitaceae family has been determined by morphology and phylogenetics, but the evolutionary relationships within the genus *Gynostemma* remain to be explored. The chloroplast genomes of seven species of the genus *Gynostemma* were sequenced and annotated, of which the genomes of *Gynostemma simplicifolium*, *Gynostemma guangxiense* and *Gynostemma laxum* were sequenced and annotated for the first time. The chloroplast genomes ranged from 157,419 bp (*Gynostemma compressum*) to 157,840 bp (*G. simplicifolium*) in length, including 133 identical genes: 87 protein-coding genes, 37 tRNA genes, eight rRNA genes and one pseudogene. Phylogenetic analysis showed that the genus *Gynostemma* is divided into three primary taxonomic clusters, which differs from the traditional morphological classification of the genus *Gynostemma* into the subgenus *Gynostemma* and *Trirostellum*. The highly variable regions of *atpH-atpL*, *rpl32-trnL*, and *ccsA-ndhD*, the repeat unilts of AAG/CTT and ATC/ATG in simple sequence repeats (SSRs) and the length of overlapping regions between *rps19* and inverted repeats(IRb) and between *ycf1* and small single-copy (SSC) were found to be consistent with the phylogeny. Observations of fruit morphology of the genus *Gynostemma* revealed that transitional state species have independent morphological characteristics, such as oblate fruit and inferior ovaries. In conclusion, both molecular and morphological results showed consistency with those of phylogenetic analysis.

## 1. Introduction

*Gynostemma* Blume is a small genus in the family Cucurbitaceae. *Gynostemma* has the ability to synthesize saponins and flavonoids. gypenoside II, IV, VIII and XII found in *Gynostemma* are homologous to ginsenosides Rb1, Rb2, Rd1 and Rf2, respectively in *Panax* spp. [1] Therefore, *Gynostemma* have been widely used in traditional medicine, mainly for the treatment of hyperlipidemia [2], diabetes [3] and inflammation [4]. To date, over 300 saponins have been isolated and identified from *Gynostemma* species [5], and the content and types of saponins, as well as their pharmacological activities, are different among species according to previous studies [6]. Thus, to make better use of *Gynostemma* plants, it is essential to fully resolve understand the taxonomic identification of this genus.

Due to hybridization, combined with facultative apomixis and polyploidy, *Gynostemma* is known for the difficulty in delimiting its species among [7]. *Gynostemma*, which comprises seventeen species and and two varieties, is native to tropical Asia and East Asia, including China, Japan, Malaysia and New Guinea [8]. China (Qinling Mountains and areas south of the Yangtze River) is the main *Gynostemma* source area, with 14 species and two varieties, including nine species and two varieties endemic to China [8,9]. The plants of this genus can be divided into two subgenera according to their fruit morphology: subgenus *Gynostemma* with globose or depressed-globose berries and subgenus *Trirostellum* with campanulate capsules [10]. Interestingly, many investigations of morphological traits have revealed transitional taxa in the wild [11]. Therefore, it is necessary to comprehensively investigate and explore the taxonomy and evolution of species within the *Gynostemma* genus at the molecular level.

*Gynostemma*, known as Jiaogulan in China, was first described as a wild edible plant in 1406 by Zhu Su in the book *Materia Medica for Famine Relief*. The genus name *Gynostemma* was first published in 1825 by German-Dutch botanist Carl Ludwig won Blume. Since then, species within *Gynostemma* have been revised many times, mostly based on morphological, ecological and cell taxonomic studies. For example, *G. pubescens* was once considered a species of the subgenus *Gynostemma*, but now, *G. pubescens* is considered a forma of *Gynostemma pentaphyllum*, not as a separate species, in the Flora of China (2011) because of the similarities in morphology [8]. Currently, DNA sequencing is a popular method for taxonomic studies, and molecular markers can provide new evidence for inter- and intra-species relationships beyond that provided by morphological traits [12]. Only a few studies have explored the phylogenetic relationships of species within *Gynostemma* based on genes or sequence fragments. By using the nuclear ribosomal internal transcribed spacer (*ITS*) [7] and chloroplast genes or fragments, Qin (*matK*, *rbcL*, and *psbA-trnH*) [11] and Abid (*ycf3*, *accD*, *petD* and *psbB*) [13], provided molecular support for the existence of subgenus units in the *Gynostemma* genus. Meanwhile, the non-monophyletic origin of *G. pentaphyllum* in the phylogenetic tree indicated the high genetic diversity of this widespread species. Compared with short DNA fragments, the complete chloroplast genome, with a length of approximately 150 kb, can provide more genetic variation information for species classification [14].

Chloroplasts contain their own genome separate from that in the cell nucleus. The chloroplast genome structure of species in different clades is relatively stable, and the genes in the chloroplast genome are fairly similar among land plants [15]. Moreover, chloroplast genomes are characterized by moderate nucleotide substitution rates compared with those of nuclear genomes. These advantages of chloroplast genomes make them an ideal tool for phylogenetic and taxonomic studies [16]. There have been multiple separate reports involving the complete chloroplast genome of a single *Gynostemma* species [17,18,19,20]. The chloroplast genomes of eight species from the *Gynostemma* genus were sequenced in Zhang’s study [21]. These studies outlined the phylogenetic relationships among *Gynostemma* species and provided valuable data for research on the relationships within *Gynostemma* species. However, for some species in subgenus *Gynostemma*, chloroplast data are still missing, and subgeneric phylogenetic relationships are also unclear.

In this study, we sequenced and assembled the complete chloroplast genomes of seven subgenus *Gynostemma* species, providing a full complement to the data on subgenus *Gynostemma* species distributed in China. We combined these newly obtained chloroplast genomes with data downloaded from GenBank to perform a comparative analysis of *Gynostemma* chloroplast genomes based on a total of 21 individuals from 14 species in this genus. This study reconstructed the phylogenetic relationships and verified the phylogenetic positions of species within the *Gynostemma* genus.

## 2. Materials and Methods

### 2.1. Plant Material, DNA Extraction, and Sequencing

Fresh, healthy leaf samples were collected from matured plants of seven species of the genus *Gynost*emma (Appendix A), with vouchers for specimens of *G. simplicifolium* placed in the Herbarium of Yunnan Branch, Institute of Medicinal Plants, Chinese Academy of Medical Sciences (IMDY), and vouchers for *G. burmanicum*, *G. caulopterum*, *G. compressum*, *G. guangxiense*, *G. laxum*, and *G. longipes*, placed in the Gynostemma Germplasm Nursery of Guangxi Medicinal Plant Garden (GXMG). Chloroplast DNA was extracted using the High Efficiency Plant Gene DNA Kit DP350 (Tiangen Biochemical Technology Co., Beijing, China), and raw reads were obtained on the Illumina NovaSeq6000 platform.

### 2.2. Chloroplast Genome Assembly and Annotation

Artificial sequences such as sequencing primers and adapters as well as some low-quality regions were removed from the raw data (Appendix A), and the quality assessment report of “CleanData” was generated using FastQC v0.11.9 and MultiQC software [22]. “CleanData” was assembled using the GetOrganelle pipeline [23] (https://github.com/Kinggerm/GetOrganelle accessed on 19 January 2022). First, 15 million clean reads were selected from the clean data to filter out possible chloroplast clean reads. Second, the SPAdes splicing program in SOAPdenovo2 [24] was used to assemble the filtered clean reads, the *G. pentaphyllum* chloroplast genome was used as the reference sequence (GenBank accession No. KX852298), and the relative positions between sequences were obtained using BLAT [25]. Finally, the full-length assembly of the sequences was performed using Bandage [26] software to obtain the full-length circular frame map of the chloroplast, and the LSC, SSC, and IRS region junctions of the full-length frame map were verified using next-generation sequencing. The assembled chloroplast genome sequences were annotated using the GeSeq program [27] and compared to the MPI-MP chloroplast gene library provided by GeSeq Nuclear coding genes (similarity = 65%) and rRNA genes (similarity = 85%) were searched for protein sequences using HMMER [28], and ARAGORN v1.2.38 [29] was used to predict tRNA genes. The annotated results were drawn using OGDRAW v 1.3.1 for the physical maps of the chloroplast genome. The other data downloaded from the NCBI (Appendix A) used in this study were reannotated using the same annotation method for the subsequent comparative analysis and phylogenetic analysis in this study.

### 2.3. Codon Usage Bias and RNA Editing Sites

The codon usage and amino acid frequency in these genes were assessed using the program codon W 1.4.4 [30]. RNA editing sites for the 10 protein-coding sequences of *Gynostemma* species were predicted by using the online PmtREP program with a cutoff value of 0.8 [31].

### 2.4. Repeat Analysis

MISA was used to identify SSRs that localizes to the chloroplast genomes of 18 species, defined for microsatellites as a unit size/minimum number of repeats ratio of 1/10, 2/6, 3/4, 4/4, 5/4, or 6/4 [32,33]. REPuter was used to identify forward, palindromic, reverse, and complementary sequences with a minimum repeat size ≥30 bp and sequence identity ≥90% (Hamming distance = 3) [34,35].

### 2.5. Comparative Analysis, and Identification of Polymorphic Loci

Alignment was visualized in Shuffle-LAGAN mode using mVISTA, using the annotated plasmid of *G. burmanicum* NC_036141.1 as a reference [36,37]. IRscope was used to analyse the contraction and expansion of the inverse repeat (IR) regions at the chloroplast genome junction [38]. DnaSP v6.12.03 software was used to calculate nucleotide variability (Pi) values and variable sites using matched chloroplast genome sequences with a window length of 600 bp and a step size of 200 bp [39].

To determine the synonymous (ks) and nonsynonymous substitution rates (ka) of 88 protein-coding genes and their ratios (ka/ks) after removing terminators from each protein gene, gene matching was performed in MEGA using MUSCLE, and the resulting data were exported from MEGA X in fasta format [30]. The ka/ks analysis was performed in DnaSP v6.12.03, and the results were interpreted as Ka/Ks > 1 for positive selection, <1 for purifying selection, and Ka/Ks = 1 for neutral selection [39].

### 2.6. Phylogenetic Analysis

To infer the evolution of the genus *Gynostemma*, we selected and downloaded chloroplast genome sequences from the NCBI for 46 species and three outgroup species (Appendix A), including *Begonia guangxiensis* (NC_046385.1), *Corynocarpus laevigatus* (NC_014807.1), and *Arabidopsis thaliana* (NC_000932.1), and a phylogenetic tree of *Cucurbitaceae* was constructed based on 86 shared protein-coding genes. We constructed a phylogenetic tree of the chloroplast genome of the genus *Gynostemma* with *Cucumis sativus* (NC_007144) as the outgroup based on the complete chloroplast genomes. All protein-coding sequences and the complete chloroplast genome sequence were aligned separately using MAFFT-v 7.490 [40]. Best-fitting models were identified for the 2 datasets using MrModeltest-v3.7 [41]. Maximum likelihood (ML) analysis was performed using the PAUP* procedure [42] with the best-fitting model GTR + G + I and 1000 bootstrap replications. Bayesian inference (BI) was achieved in Mrbayes-v3.2 [43] with the GTR + G + I model. The parameters used are: Markov chain Monte Carlo (MCMC) algorithm run for 1 × 108 generations and extracted once every 1000 generations. The first 25% of samples were discarded as burn-in. Stationarity was considered to be achieved when the average standard deviation of the splitting frequencies remained below 0.001. The phylogenetic trees were visualized by FigTree-v1.4.4. The topological structures of the two phylogenetic trees were consistent, so they were manually combined using AI software.

## 3. Results

### 3.1. Characterization of the Chloroplast Genomes of the Gynostemma Genus

To gain some insight into the phylogenetic positions of species within *Gynostemma*, it is important to obtain chloroplast genome data for as many species as possible in the genus under study. To this end, seven *Gynostemma* species, namely, *G. burmanicum*, *G. caulopterum*, *G. compressum*, *G. guangxiense*, *G. laxum*, *G. longipes*, and *G. simplicifolium*, were sampled for the chloroplast genome sequencing. The chloroplast genomes ranged in size from 157,419 bp to 157,840 bp (Appendix A). The chloroplast genomes of all *Gynostemma* species are typically quadripartite in structure with a large single-copy region (LSR: 85,831–86,470 bp), a small single-copy regions (SSR: 18,510–18,636 bp) and two inverted repeat regions (IRA and IRB: 26,114–26,295 bp) (Figure 1). The GC contents of the chloroplast genomes of all seven species were quite similar (36.96–36.98%), of which the GC contents of the IR, LSC and SSC regions were 42.74–42.80%, 34.75–34.87% and 30.64–30.84%, respectively (Appendix A). This high GC percentage in the IR regions was caused by the rRNA genes distributed in these regions. A total of 133 genes were identified in each chloroplast genome, including 87 protein-coding genes, eight rRNA genes, 37 tRNA genes and one pseudogene (Appendix A). These genes were classified into three groups according to their functions, including photosynthesis, self-replication and others (Table 1). The gene distributions in these seven chloroplasts were the same: the LSC and SSC regions encoded 83 and 12 genes, respectively, and the IR regions contained 19 duplicate genes. There were 22 intron-containing genes, of which 20 genes had one intron and two genes had two introns (Table 1). The basic information and gene contents of 14 previously reported chloroplast genomes of *Gynostemma* species are presented in Appendix A. By comparing all sequenced chloroplast genomes generated in this study with 14 other previously reported chloroplast genomes of *Gynostemma* species (Appendix A), we found that they had a highly conserved gene content, gene number, orientation and intron number.

### 3.2. Analysis of Codon Usage Bias and Prediction of RNA Editing Sites

The relative synonymous codon usage (RSCU) value was calculated to show the codon usage frequency based on protein-coding genes of seven *Gynostemma* chloroplast genomes. The chloroplast genome protein-coding genes of each *Gynostemma* species were composed of 26,614–26,723 codons. Except for methionine (Met) and tryptophan (Trp), which were encoded by a single codon, the amino acids were encoded by two to six synonymous codons and displayed a preference for certain codons. Among these amino acids, leucine (Leu, 10.5%) was the most abundant amino acid, whereas cysteine (Cys, 1.2%) was the least universal amino acid in these chloroplast genomes. The RSCU values of all codons are shown in Figure 2 (Appendix A). Approximately half of the codons were used more frequently, with RSCU values greater than 1 (32/64), and almost all biased codons ended with A/U (29/32). The most and least commomly used codons were AUU (M), encoding leucine, and UGC (N), encoding cysteine, respectively. According to the codon usage bias analysis, the RSCU values were very similar in all *Gynostemma* chloroplast genomes, and the frequency of different codons coding for the same amino acid was almost the same in all *Gynostemma* species.

RNA editing is an important post-transcriptional modification process and has been observed in many published chloroplast genomes. To reveal the composition and characteristic of RNA editing of the *Gynostemma* chloroplast genome, RNA editing sites in seven newly sequenced chloroplast genomes were predicted. For each chloroplast genome, approximately 15–17 RNA editing sites distributed in 10 protein-coding genes were predicted (Appendix A). The RNA editing site in the *rps11* gene (nucleotide position 91) was predicted from the chloroplast genomes of *G. burmanicum*, *G. caulopterum*, *G. compressum*, *G. guangxiense*, and *G. longipes*, but was not present in the *G. laxum* and *G. simplicfolium* chloroplast genomes. Three interspecies differential RNA editing sites were predicted in the *rps4* gene of the *Gynostemma* chloroplast genome. In *G. burmanicum*, *G. guangxiense*, *G. laxum* and *G. simplicifolium*, the editing site occurred at nucleotide position 602, causing ACT(T) to ATT(I) conversion, whereas in *G. caulopterum*, *G. compressum* and *G. longipes*, the editing sites occurred at the nucleotide positions 496 and 503 causing CTT (L) to TTT (F) and CCA (P) to CTA (L) conversions, respectively. Unfortunately, these interspecies differences in RNA editing sites did not exhibit clear subgenus specificity. In addition, two identified RNA editing sites were found at the start codons of *psbL* and *ndhD*.

### 3.3. Repeat Analysis

The characteristics of SSR copy number polymorphism make it a valuable molecular metric for genetic diversity research and evolution research. The repeat analysis of chloroplast SSRs plays a crucial role in taxonomic and phylogenetic studies of plant species. MISA was used to identify SSRs of 21 *Gynostemma* chloroplast genomes (Figure 3A,B). For the seven newly sequenced chloroplast genomes involved in this study, we identified 64, 67, 57, 59, 59, 71, and 56 SSRs in *G. burmanicum*, *G. caulopterum*, *G. compressum*, *G. guangxiense*, *G. laxum*, *G. longipes*, and *G. simplicifolium*, respectively. Mononucleotide, dinucleotide, trinucleotide, and tetranucleotide repeat units were identified in all *Gynostemma* species, but no pentanucleotide repeat units or hexanucleotide repeat units were found (Figure 3B). The A/T and AT/TA repeat units were the most abundant mononucleotide and dinucleotide types, respectively, in all *Gynostemma* chloroplast genomes, accounting for approximately 81–87% of the total number of SSRs. In contrast, C/G repeats were very rare. This result is consistent with the phenomenon that most abundant SSRs consist of polyA or polyT repeats in most chloroplast genomes [44]. The composition of trinucleotide repeats varied among different species, with *G. guangxiense*, *G. compressum*, *G. caulopterum*, and *G. longipes* having one ATC/ATG motif and no AAG/CTT motifs, and *G. laxum*, *G. burmanicum*, and *G. simplicifolium* having an AAG/CTT motif and no ATC/ATG motifs (Figure 3A). Most of the identified SSRs were within the intergenic region (IGS), while fewer SSRs were located in the intron region or the protein-coding region (CDS) (Appendix A).

REputer, another repeat analysis tool, was also used to detect four types of long repeat sequences in the *Gynostemma* chloroplast genomes, including forward, palindromic, reverse, and complementary repeats (Figure 3C,D), which are thought to play an important role in genome rearrangements. There are certain differences in the types and numbers of repeats in chloroplast genomes of different species of *Gynostemma*. For all *Gynostemma* chloroplast genomes, forward and palindromic repeats were the most common repeat types. Only zero to four reverse or complementary repeats are present in most *Gynostemma* chloroplast genomes. Compared with the chloroplast of most species in this genus, which contains approximately 40 repeats, the *G. caulopterum* chloroplast contains 90 repeats, more than twice as many as the numbers of other species. The size of repeats varies among 21 species, and most of the repeats exist in the range of 35–34 bp (Figure 3D). *G. caulopterum* chloroplast contains more long length repeats (>45 bp) than other species chloroplasts. These SSRs and the long repeats identified in the *Gynostemma* chloroplast genome can be used as significant molecular markers to explore the genetic diversity and phylogeny of *Gynostemma* in future studies.

### 3.4. Comparative Analysis and Selection Pressure Analysis

To elucidate the chloroplast genome divergence of different *Gynostemma* species, we used mVISTA to perform a comparative analysis based on all available chloroplast genome data for this genus (14 samples downloaded from the NCBI and seven newly sequenced samples in this study) (Figure 4). The comparison revealed that the chloroplast genomes of different species of *Gynostemma* were highly similar. The IR regions were less divergent than the LSC and SSC regions. Furthermore, it was observed that protein-coding regions showed higher conservation than noncoding regions. The greatest divergence was found in intergenic regions, including *trnK*-*rps16*, *trnS-trnG*, *trnR-atpA*, *atpH-atpI*, *rpoB-trnC*, *petN-psbM*, *trnT-psbD*, *psaA-pafI*, *trnT-trnL*, *trnF-ndhJ*, *ndhC-trnV*, *rpl32-trnL*, *ccsA-ndhD* and *trnH-psbA*. In addition, high sequence divergence was found only in three protein-coding regions, *rps16*, *ndhF* and *ycf1*.

To quantify differences in chloroplast genomes from different *Gynostemma* species, the DNA polymorphism analysis was performed to detect highly variable sites by calculating the nucleotide diversity (Pi) value (Figure 5). The average Pi value was 0.0072. The IR regions showed much lower variability in Pi values than the LSC and SSC regions. Pi values higher than twice the median (Pi > 0.0124) were used to identify mutational hotspots. The region that showed the highest Pi value was *petN-psbM* (Pi ~ 0.04), and the top 10 regions with pi values are marked, included nine intergenic regions and one protein-coding region. Due to their high variability, these regions can be used as candidate molecular markers for plant identification and phylogenetic analysis in *Gynostemma*.

The ratio of synonymous substitutions (Ks) to nonsynonymous substitutions (Ka) is an important reference for determining whether a mutation is neutral, detrimental, or beneficial, with Ka/Ks > 1 indicating a beneficial mutation, Ka/Ks < 1 indicating a detrimental mutation, and Ka/Ks = 1 indicating a neutral mutation. We compared the chloroplast genomes of 21 individuals of the genus *Gynostemma* to calculate their Ks, Ka, and Ka/Ks (Appendix A). We calculated Ks and Ka and their ratios for a total of 87 genes, 10 of which could not be determined due to a lack of information (Ks = 0). After removing these 10 genes, a total of 10 genes had Ka/Ks > 0.5, of which only rpl14 had Ka/Ks > 1 and was positively selected.

### 3.5. IR Region Contraction and Expansion

The IR region is considered as the most conserved region in the chloroplast genome. The variation in chloroplast genome size is generally thought to be caused by expansion/contraction of the IR region. The IR expansion/contraction of *Gynostemma* chloroplast genomes was analysed by performing collinearity analysis of the genomes (Appendix A) and comparing the boundary structure of LSC, SSC, and IRs regions among the *Gynostemma* chloroplast genomes (Figure 6). The collinearity analysis was conducted by using MAUVE. The entire chloroplast genome sequence was a homologous region without large fragment rearrangements or loss. The perfect collinearity indicated that the chloroplast genomes within this genus were relatively conserved. In all species, *rps19* flanks the LSC/IRb junction, with part of the sequence present in the IR and the rest in the LSC. Although the length of the *rps19* gene in IRb varied among species, the results did not suggest subgenus specificity. For the widespread species *G. pentaphyllum*, the length of the *rps19* gene to the LSC/IRb boundary was different among samples (2 bp, 20 bp and 51 bp). Except in *G. microspermum* and *G. burmanicum*, the *ycf1* fragment and *ndhF* genes were located at the boundary of SSC/IRb. In both *G. microspermum* and *G. burmanicum*, the *ndhF* gene was not situated at the SSC/IRb boundary; however, the reasons were different. In *G. microspermum*, which has the shortest IR region length in the *Gynostemma* genus, the contraction of the IR region caused the loss of approximately 60 bp of the *ycf1* fragment and the *ndhF* gene to move away from the SSC/IRb boundary. In our newly sequenced *G. burmanicum*, the frameshift mutation and premature termination caused by the 2 bp deletion of the *ndhF* gene resulted in the absence of the *ndhF* gene at the SSC/IRb boundary. However, this deletion mutation was not observed in the other two published samples of *G. burmanicum*. In all *Gynostemma* species, the *ycf1* genes were located at the boundary of SSC/IRa. The length of the overlapping regions of the *ycf1* gene and SSC showed clear subgenus differences. The overlapping region of all *Gynostemma* subgenus species is 4499 bp in length, while that of *Trirostellum* subgenus species is slightly longer ranging from 4508 to 4653 bp. No gene stretches across the boundary between the LSC and IRa regions of all *Gynostemma* species. The *trnH* gene is 25–44 bp away from the LSC/IRa boundary. Interestingly, the length from the *trnH* gene to the LSC/IRa boundary was different among different individuals of *G. burmanicum*, *G. pentaphyllum* and *G. longipes*. Overall, the IR boundaries in the nine *Gynostemma* species showed similar characteristics with only slight differences in the flanking region or distance from the boundary in the organized genes, including *rps19*, *ycf1*, *ndhF* and *trnH*.

### 3.6. Phylogenetic Relationships among the Gynostemma Species

At present, although the phylogenetic tree shows that *Gynostemma* belongs to the basal group of Cucurbitaceae (Appendix A), the phylogenetic relationships of species within this genus are still controversial. According to the Flora of China (2011) [8], two subgenera, *Gynostemma* (eight species and two variants) and *Trirostellum* (six species), are included within this genus. To obtain better resolution of the phylogenetic relationships within the *Gynostemma* genus, we constructed a phylogenetic tree using 21 whole chloroplast genome sequences from nine subgenus *Gynostemma* and five subgenus *Trirostellum* species. To explore the taxonomic relationships of different populations within a species, multiple sequencing datasets of the same species were used for phylogenetic analysis. The Bayesian inference (BI) and maximum likelihood (ML) analyses yielded phylogenetic relationships with the same topology. The phylogenetic tree with supporting values is presented in Figure 7 *Gynostemma* formed four clades, supported by Bayesian posterior probabilities of 1 and maximum likelihood bootstrap support of 100%. Clade I included *G. microspermum* of subgenus *Trirostellum*, which occupied the most basal position in this genus. The three other species of subgenus *Trirostellum* (*G. cardiospermum*, *G. laxiflorum* and *G. yixingense*) clustered on one branch as Clade II, the closest clade to Clade I. The members of Clade IV, the clade farthest from the base, contained species all belonging to subgenus *Gynostemma*, including *G. laxum*, *G. pubescens*, *G. simplicifolium*, *G. burmanicum*, *G. pentaphyllum*, and *G. longipes*. Clade III appeared to be a transitional clade whose members included *G. caulopterum*, *G. longipes*, *G. compressum*, *G. pentaphyllum*, and *G. guangxiense* of subgenus *Gynostemma* and *G. pentagynum* of subgenus *Trirostellum*. Interestingly, different samples of the widespread species *G. pentaphyllum* and *G. longipes* existed in both Clades III and IV.

### 3.7. Morphological Analysis

The classification within the genus *Gynostemma* is mainly based on the morphological characteristics of the fruits. Here, we summarize the morphological characteristics of *Gynostemma* species through literature review, specimen observation and field investigation (Table 2). Obviously, in the taxonomic literature of *Gynostemma*, the fruit types of *G. pentagynum*, *G. guangxiense*, *G. compressum* and *G. caulopterum* are confused or not indicated. According to our field investigation, *G. caulopterum*, *G. guangxiense* and *G. compressum* from the subgenus *Gynostemma* and *G. pentagynum* from the subgenus *Trirostellum* share some common morphological characteristics, such as oblate fruit shape (2–5 ribs), persistent perianth and style, and inferior ovary. The typical subgenus *Gynostemma* fruits are globose berries, while typical subgenus *Trirostellum* fruits are campanulate capsules. Therefore, the fruit shapes of oblate distinguishing the group from the rest of the species are shared by this group. The genus *Gynostemma* is divided into two subgenera in traditional morphology *Gynostemma* and *Trirostellum*, but phylogenetic analysis shows that the genus *Gynostemma* is mainly divided into three independent branches. Therefore, we propose the existence of intermediate transitional branches. The typical subgenus *Gynostemma* fruits are globose berries, while the typical subgenus *Trirostellum* fruits are campanulate capsules. The species of Clade III, *G. pentagynum*, *G. guangxiense*, *G. compressum* and *G. caulopterum*, share a particular fruit shape, the oblate fruit, which is different from both of the subgenera *Gynostemma* and *Trirostellum* (Figure 8). Interestingly, the species in the transitional branch happen to be those with oblate fruit morphology. The perfect correspondence between the three fruit shapes and the phylogenetic tree suggested possible subgenus taxon reformation.

## 4. Discussion

### 4.1. General Characteristics of the Chloroplast Genomes of the Genus Gynostemma

*Gynostemma* is an economic plant genus in Southeast Asia, with saponin components that vary among species or populations [5,47]. Obtaining information about the intragenus and infraspecific variation of the chloroplast genomes is an important step in genetic research on this genus and a basis for the development and application of *Gynostemma* resources [48]. To date, comparative analyses of complete chloroplast genomes have made significant contributions to reconstructing phylogenies at different taxonomic levels in plants, including species of the genus *Gynostemma* [21,49]. However, in lower taxonomic units, for example, at the intraspecies or within-genus level, there is probably less variation in the chloroplast genome, which is mainly observed in hotspot regions [50].

In this study, we sequenced and assembled the complete chloroplast genome of seven *Gynostemma* species, three of which were reported for the first time, including those of *G. simplicifolium*, *G. laxum* and *G. guangxiense*. Furthermore, we downloaded all the available chloroplast genomes for this genus and performed a comprehensive analysis of the chloroplast genomes of this basal group of Cucurbitaceae. The chloroplast genomes of different species of *Gynostemma* are highly conserved, and the gene compositions, structures and GC contents are similar, RNA editing sites and condon usage bias showing close species relationships within the genus *Gynostemma* [13,17,18,19,20,21,51]. Our newly sequenced chloroplast genomes of *Gynostemma* species revealed differences between the two subgenera, despite the low chloroplast polymorphism in this genus.

### 4.2. Analysis of Codon Usage Bias and Prediction of RNA Editing Sites

Codon usage bias is closely associated with gene expression. Therefore, the origin, mutation and evolutionary patterns of species or genes are usually reflected by the use of codons. The most enriched amino acid within the *Gynostemma* species was leucine, and this result was frequently reported in other angiosperms [52]. The frequently used codons (RSCU > 1) usually end in A/U, which is consistent with reports from other plants [51]. The occurrence of base addition, loss, or conversion in the coding region after transcription is known as RNA editing. C-to-U RNA editing occurs mostly in angiosperms [53]. Two RNA editing events were detected in this study. *psbL* and *ndhD* could be translated normally because of the conversion of ACG (Thr) to the start codon AUG. The unconventional codon phenomenon in *psbL* and *ndhD* is common in *Cucurbitaceae* [54] and is also present in plants such as *Betula platyphylla* [55] and *Nicotiana tabacum* [53].

### 4.3. Phylogenetic Relationships

At present, phylogenetic relationships and interspecific taxonomic relationships can be demonstrated by complete chloroplast genomes and protein-coding genes based on a large number of literature reports [54,56]. The phylogenetic tree of *Cucurbitaceae* was established using the protein CDS region, and this phylogenetic tree indicated with a high bootstrap value that the genus *Gynostemma* is clearly related among the genera of *Cucurbitaceae*, which is consistent with the results of Zhang’s study [57]. The phylogenetic tree shows that the genera *Gerrardanthus*, *Cyclantheropsis*, *Hemsleya* and *Gynostemma* constitute the earliest divergent lineages of *Cucurbitaceae*, which is consistent with the APG IV classification system [58]. As in previous studies [18], the phylogenetic tree within the genus *Gynostemma* indicated that *G. microspermum* was the earliest divergent lineage of *Gynostemma*. In addition, two interesting phenomena occurred in the third branch: First, in contrast to previous studies, *G. pentaphyllum* and *G. longipes* were found to occur in the Clade III, which may be due to the hybridization and maternal inheritance of plastids [59]. The phylogenetic tree constructed by gene fragments similarly indicated that *G. pentaphyllum* is nonmonophyletic in origin [11,13]. Second, the phylogenetic tree of the genus *Gynostemma* is divided into three primary taxonomic clusters, which differs from the traditional morphological classification of the genus *Gynostemma* into the subgenus *Gynostemma* and *Trirostellum* [8,10]. Clade III has species of both subgenus *Trirostellum* and subgenus *Gynostemma*, defined as a transitional branch. Studies on the microstructural characteristics of the seed coat of the genus Gynostemma showed the existence of transitional forms between the two subgenera [60], and a strict concordance tree was obtained based on the sequences of the *ITS*, *matK*, and *rpcL* genes of Gynostemma, which also revealed that the two genera did not form two independent branches [7]. Evolutionary correlations within the genus *Gynostemma* are better revealed by a sufficient sample sizes of chloroplast genomes [61].

### 4.4. Identification of Suitable Polymorphic Loci at the Subgenus and Species Levels

The chloroplast genome is abundant in phylogenetic information [62], and visualization of this information in this research revealed gene fragments with consistent patterns of phylogenetic relationships with *Gynostemma*. The fragments obtained from the mVISTA analysis are *trnR-atpA*, *atpH-atpL*, *rpoB-psbD*, *psbZ-rps14*, *trnP-psaJ*, *rpl32-trnL*, *ccsA-ndhD*, and *trnV-ocf70*, of which *atpH-atpL*, *rpl32-trnL*, and *ccsA-ndhD* are polymorphic regions. These hypervariable polymorphic regions are considered to provide valuable markers for elucidating phylogenetic relationships among *Gynostemma* species. Microsatellites (SSRs) are widely present in plant chloroplast genomes and have properties such as polymorphism, codominant inheritance and multiallelism [63]. Therefore, SSR molecular markers are often applied in genotype identification, genetic analysis, population structure detection and biogeographic analysis [64,65]. When we investigated the SSRs of the chloroplast genome of the genus *Gynostemma*, interestingly, we found that trinucleotide repeat units were associated with phylogenetic position. Clades I and II lacked AAG/CTT and ATC/ATG repeats; Clade III included ATC/ATG repeats but lacked AAG/CTT repeats except in *G. pentagynum*; and Clade IV included AAG/CTT repeats but lacked ATC/ATG repeats. The pattern was basically consistent with the pattern of phylogenetic tree branching. This consistency is not a coincidence, instead suggesting the important role of SSRs for gene rearrangement and phylogenetic studies. In addition, analysis of the IR boundaries revealed that the lengths of the overlapping regions of *rps19* and IRb and of *ycf1* and SSC have clear subgenus differences. Polymorphic loci in the chloroplast genomes at the subgenus and species levels enable clear visualization of the evolutionary process of the species. For example, *G. microspermum* forms an independent branch in the most basal position of the genus *Gynostemma*, and its IR contraction and expansion are different from those of other species. *G. pentagynum* occupies the most basal position of Clade III, and its polymorphic fragments and SSR fragment variation regularity are more similar to those of Clade II.

### 4.5. Association of Fruit Shapes with the Phylogenetic Tree

Based on extensive field observations of *Gynostemma* fruit morphology and a comprehensive review of the *Gynostemma* taxonomic literature (Figure 8), we found that the morphology-based classification supported the chloroplast genome-based phylogenetic tree well. In other words, there may be a transitional group between the subgenus *Gynostemma* and *Trirostellum*. In summary, our study is the most comprehensive taxonomic and molecular evolutionary study of the *Gynostemma* genus based on chloroplast genomes.

**Figure 8 genes-14-00929-f008:**
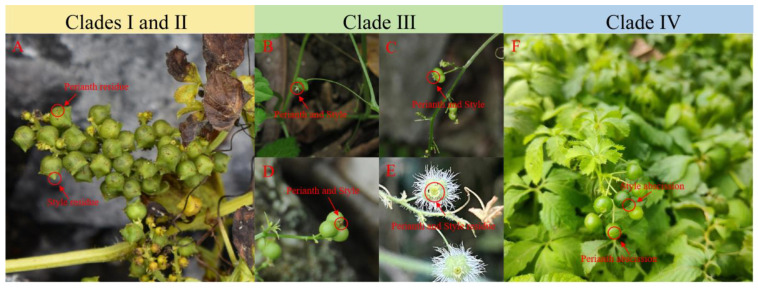
Fruit morphology of some *Gynostemma* species. (**A**): *G. microspermum* (capsule with a persistent perianth and style); (**B**): *G. compressum*, (**C**): *G. caulopterum*, (**D**): *G. guangxiense* and (**E**): *G. pentagynum* (oblate fruit with an inferior ovary and a persistent perianth and style); (**F**): *G.longipes* (berry without a persistent perianth and style).

## 5. Conclusions

In this study, the seven chloroplast genomes of the genus *Gynostemma* were sequenced, assembled, analyzed, and compared with 14 other published chloroplast genomes of *Gynostemma* species. Phylogenetic analysis was performed to clarify the phylogenetic relationships within the genus *Gynostemma*. The taxonomic relationships of subgenera were inferred from the results of the phylogenetic tree, chloroplast genome fragments, and morphology. The mVISTA, SSR and IR boundary results revealed differences at the subgenus level. By discovering the correlation between fruit morphology and the phylogenetic tree, a new hypothesis for the subgenus level classification of *Gynostemma* was proposed. In addition, molecular markers for highly polymorphic regions at the subgenus level were provided for further taxonomic and DNA barcode studies.

## Figures and Tables

**Figure 1 genes-14-00929-f001:**
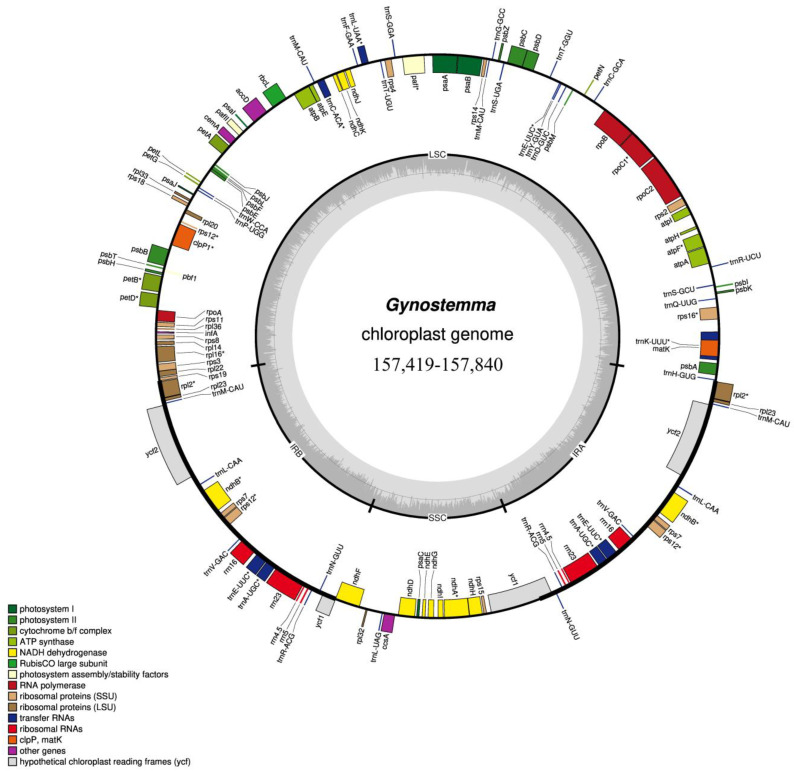
Chloroplast genome maps of *Gynostemma*. Genes transcribed counterclockwise are present inside of the circle. Genes transcribed clockwise are present outside of the circle. The colour of the genes corresponds to the function of the genes. The dashed area of the inner circle indicates the GC content of the genome. * Genes with intron.

**Figure 2 genes-14-00929-f002:**
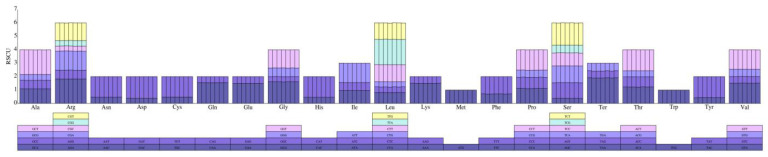
Comparative analysis plots of RSCU values of seven species in the genus *Gynostemma*. The bar chart above each amino acid shows RSCU values within *Gynostemma* species. Each bar represents a species, from left to right: *G. burmanicum*, *G. caulopterum*, *G. compressum*, *G. guangxiense*, *G. laxum*, *G. longipe*s and *G. simplicifolium*.

**Figure 3 genes-14-00929-f003:**
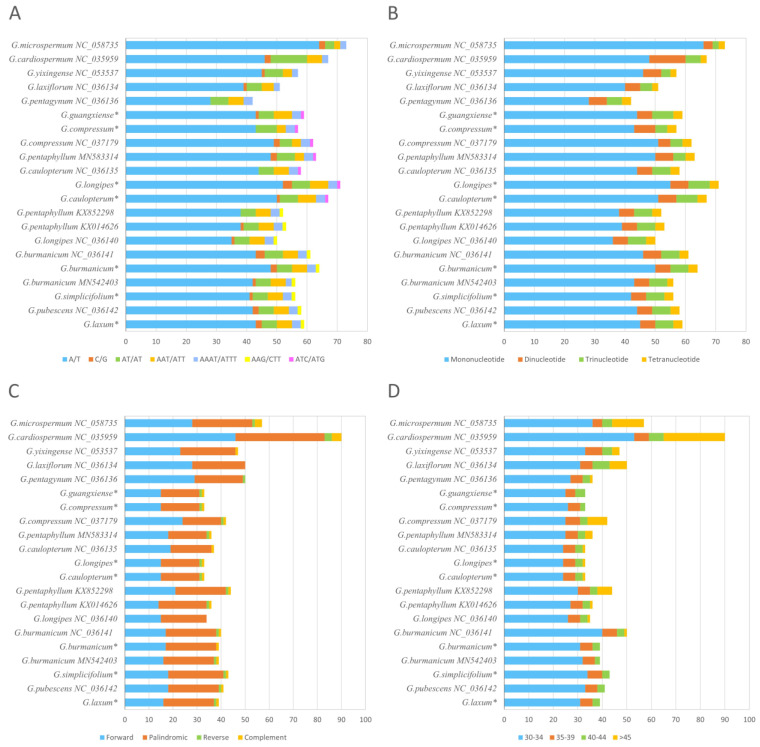
Repeat sequence signature map. (**A**): SSRs for repeated unit classification. (**B**): comparison of mononucleotide to tetranucleotide repeat sequence types. (**C**): palindromic, forward, reverse, and complementary repeat sequence types; (**D**): Repeat length classification of long repeat sequences. * Represents a newly sequenced species of the genus *Gynostemma*.

**Figure 4 genes-14-00929-f004:**
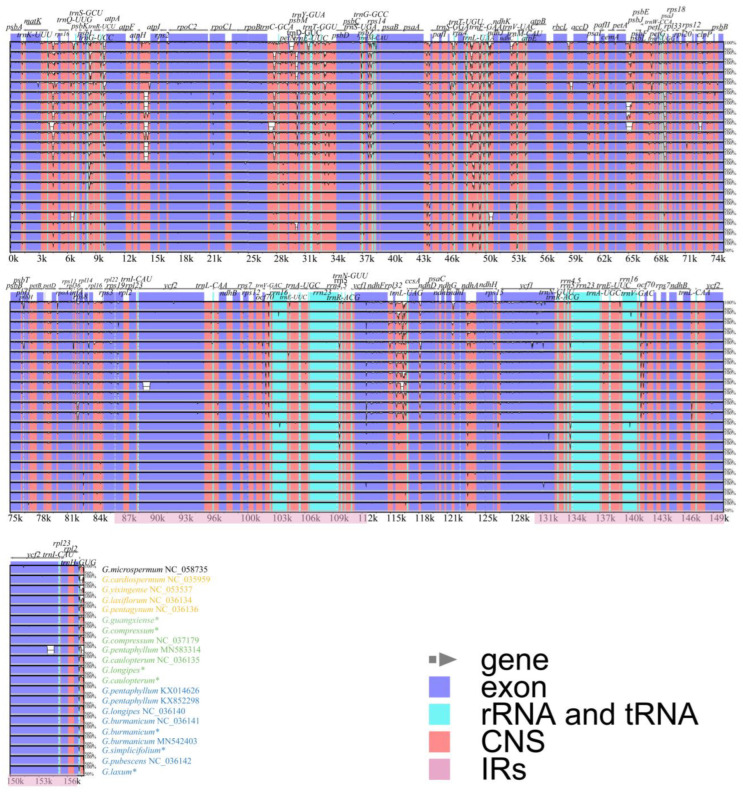
Comparison of the chloroplast genomes of *Gynostemma*. Global Shuffle-LAGAN alignment was performed on the mVISTA website. Grey arrows above the alignment represent the genes. In each plot, the vertical scale represents the percent identity (50 to 100%). Genome regions are colour-coded as protein-coding exon (blue), rRNA or tRNA (sky blue), and conserved noncoding sequences (CNS, red). * Represents a newly sequenced chloroplast genome of *Gynostemma* species in this study.

**Figure 5 genes-14-00929-f005:**
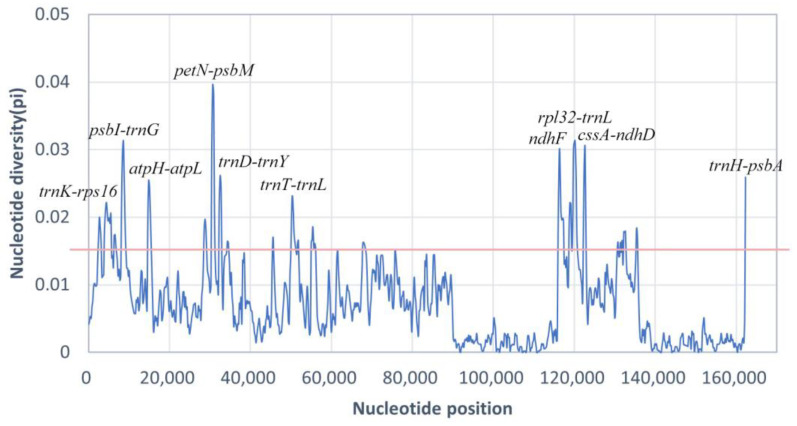
Comparative analysis of nucleotide diversity (Pi) values among chloroplast genomes of *Gynostemma* species. The pink line is twice the median Pi value (Pi = 0.0124).

**Figure 6 genes-14-00929-f006:**
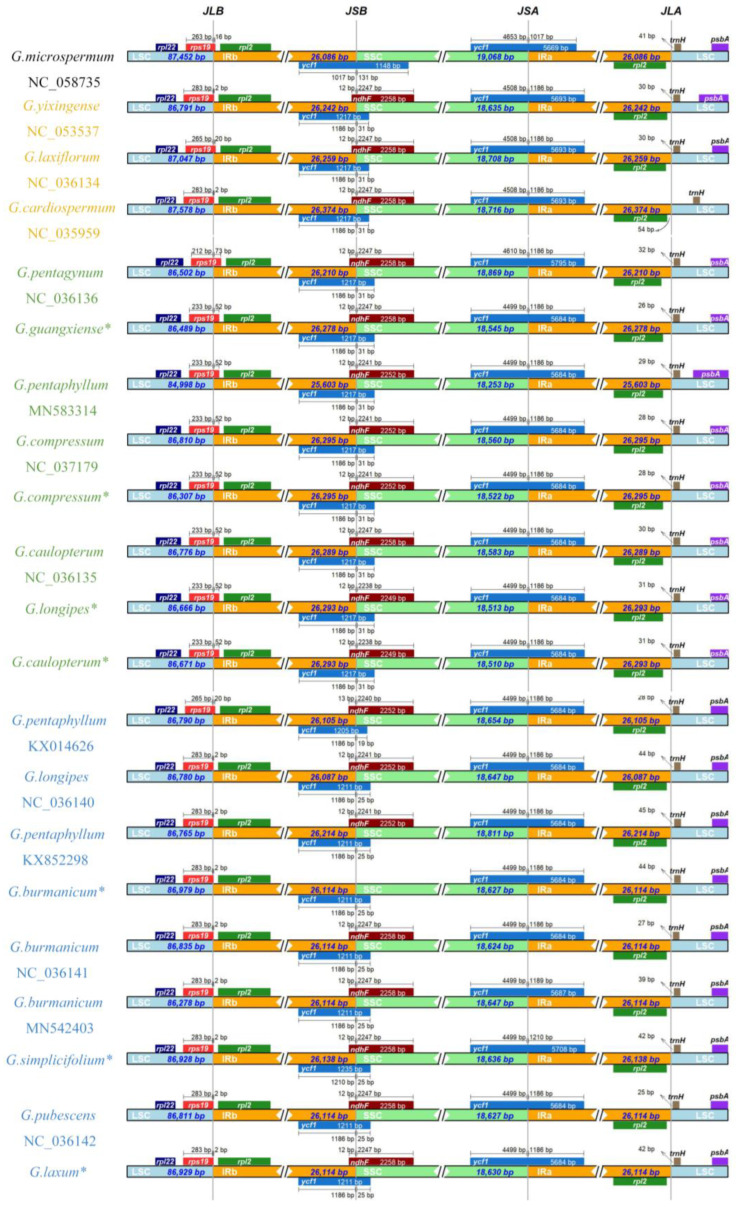
Comparison of SC/IR junctions among the chloroplast genomes of *Gynostemma* species. JLB (junction of LSC/IRb), JSB (junction of SSC/IRb), JSA (junction of SSC/IRa), JLA (junction of LSC/IRa). * Represents newly sequenced chloroplast genomes of *Gynostemma* species in this study.

**Figure 7 genes-14-00929-f007:**
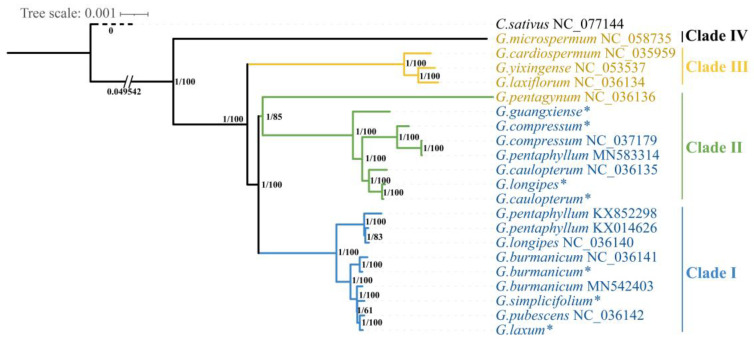
Phylogenetic tree of complete chloroplast genomes of the genus *Gynostemma* based on maximum likelihood (ML) and Bayesian inference (BI) methods. The Bayesian posterior probabilities/ML bootstrap support values are displayed at the nodes. The colour of the species name indicates the subgenus: blue—subgenus *Gynostemma*, yellow—subgenus *Trirostellum*. branch length representative substitutions per site. * Represents newly sequenced chloroplast genomes of *Gynostemma* species in this study.

**Table 1 genes-14-00929-t001:** Genes encoded in common among seven species of the genus *Gynostemma*.

Gene Functions	Biological Function	Gene List	Number
Photosynthesis	ATP synthase related	*atpA atpF atpH atpI atpE atpB*	6
	Photosystem I	*psaB psaA psaI psaJ psaC*	5
	Photosystem II	*psbA psbK psbI psbM psbD psbC psbZ psbJ psbL* *psbF psbEpsbB psbT psbH*	14
	Cytochrome b/f complex	*petN petA petG petL petB * petD **	6
	NADH dehydrogenase	*ndhJ ndhK ndhC ndhB * ndhF ndhD ndhE ndhG* *ndhI ndhA * ndhH ndhB **	12
	Photosystem biogenesis factor 1	*pbf1*	1
	Photosystem I assembly protein	*pafI ** pafII*	2
Self-replication	Ribosomal Structural RNAs	*rrn16 rrn23 rrn4.5 rrn5 rrn5 rrn4.5 rrn23 rrn16*	8
	Translation-related gene	*trnK-UUU * trnQ-UUG trnS-GCU trnR-UCU* *trnC-GCA trnD-GUC trnY-GUA trnE-UUC* *trnT-GGU trnS-UGA trnG-GCC trnfM-CAU* *trnG-UCC trnS-GGA trnT-UGU trnL-UAA ** *trnF-GAA trnV-UAC * trnM-CAU trnW-CCA* *trnP-UGG trnl-CAU trnL-CAA trnV-GAC* *trnE-UUC * trnA-UGC * trnR-ACG trnN-GUU* *trnL-UAG trnN-GUU trnR-ACG trnA-UGC ** *trnE-UUC * trnV-GAC trnL-CAA trnl-CAU* *trnH-GUG*	37
	Ribosomal Proteins (small subunit)	*rps16 * rps2 rps14 rps4 rps18 rps12 * rps12 * rps11 rps8 rps3 rps19 rps7 rps15 rps7*	14
	Ribosomal Proteins (large subunit)	*rpl33 rpl20 rpl36 rpl14 rpl16 * rpl22 rpl2 * rpl23 rpl32 rpl23 rpl2 **	11
	RNA polymerase	*rpoC2 rpoC1 rpoB rpoA*	4
Other genes	RuBisCO large subunit	*rbcL*	1
	Translation related	*infA*	1
	Acetyl-CoA carboxylase gene	*accD*	1
	RNA Splicing	*matK*	1
	Carbon metabolism	*cemA*	1
	c-type Cytochrome biogenesis	*ccsA*	1
	ATP-dependent protease subunit	*clpP1 ***	1
	Unknown	*ycf2 orf70 ycf1 ycf1 orf70 ycf2*	6

* Genes with one intron. ** Genes with two introns.

**Table 2 genes-14-00929-t002:** *Gynostemma* species information summary.

Subgenus	Species	Fruit			Persistence		Information Source
		Type	Mature Performance	Shape	Perianth	Style	
*Trirostellum* [10]	*G. microspermum*	capsule	split	campanulate	-	Yes	[8] and Substance
	* G. cardiospermum *	capsule	split	campanulate	-	Yes	[8]
	* G. yixingense *	capsule	split	campanulate	-	Yes	[8]
	* G. laxiflorum *	capsule	split	campanulate	-	Yes	[8]
	* G. pentagynum *	-	-	5-angled-oblate	Yes	Yes	[8] and Substance
*Gynostemma* [10]	* G. guangxiense *	-	-	3-angled-oblate	Yes	Yes	[8,45] and Substance
	* G. compressum *	-	-	compressed, obtriangular	Yes	Yes	[8] and Substance
	* G. caulopterum *	-	-	compressed globose	Yes	Yes	[8,46] and Substance
	* G. pentaphyllum *	berry	not split	globose	-	-	[8]
	* G. longipes *	berry	not split	globose	-	-	[8]
	* G. burmanicum *	berry	not split	globose	-	-	[8]
	* G. simplicifolium *	berry	not split	globose	-	-	[8]
	* G. laxum *	berry	not split	globose	-	-	[8]

Species font color corresponds to the clade of phylogenetic tree.

## Data Availability

This seven complete chloroplast genomes have been deposited at GenBank with accession numbers ON872370 (*G. burmanicum*), ON872371 (*G. caulopterum*), ON872372 (*G. compressum*), ON872373 (*G. guangxiense*), ON872374 (*G. laxum*), ON872375 (*G. longipes*) and ON872376 (*G. simplicifolium*).

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
