# Peer review of "The Complete Chloroplast Genomes of Gynostemma Reveal the Phylogenetic Relationships of Species within the Genus"

_genes, 2023, doi:10.3390/genes14040929_

Round 1

Reviewer 1 Report

Authors describe a research work on Gynostemma by resorting to chloroplast sequencing and tried to resolve the phylogentic relationship among the different sub-genera or species of Gynostemma utilizing bioinformatics approaches.  This research work was well planned and results are clearly presented in the manuscript. Overall the manuscript is drafted well. However, some concerns in the present version of the ms are as below:

Despite a prior publication (https://www.frontiersin.org/articles/10.3389/fpls.2017.01583/full) describing phylogentics of Gynostemma based on chloroplast genomic approaches authors have undertaken this study. The rationale for the same should be clearly explained in the introductory paragraphs. Authors should also  discuss their results by comparing similar studies in other plants (preferably with any of the closest genera of Gynostemma) where chloroplast genome sequences have been utilized to resolve the phylogenetic relationships. It is completely missing in the discussion. Discussion pertaining to some of the important publications are also missing. (for eg. Abid et al., 2019 https://www.sciencedirect.com/science/article/abs/pii/S0367326X19314157)

Minor concerns:

Many places (including figures and tables) the scientific names are not italicized, gene names are not italicized. Appropriate conventions shall be followed. 

Cite the following articles too:

Zhao, Y., Zhang, X., Zhou, T. et al. Braz. J. Bot (2023). https://doi.org/10.1007/s40415-023-00874-z

Zhang, P., Xu, W., Lu, X. et al.  Physiol Mol Biol Plants 27, 2727–2737 (2021). https://doi.org/10.1007/s12298-021-01105-z

Abid et al., (2019) https://doi.org/10.1016/j.fitote.2019.104295

Reviewer 2 Report

- The manuscript is well written but lacks some of the analyzes that are expected in this type of work (for example signal selection and prediction of RNA editing sites). Programs like DNAsp, palm, and Prepact can be used for this, respectively.

- Although the authors present the NCBI accession numbers, when searching for them there is no return. It is important that data is available as soon as possible.

- I honestly could not understand the connection between the fruit morphology results with the rest of the manuscript. I found it disjointed and suggest removing that part or explaining it better.

- Figures 3 and 4 are below the expected quality standard for publication (many unnecessary lines, species names not italicized, colors not very striking, and parts A, B, C, and D centralized). I believe that it is possible to greatly improve the quality of the figures in a future version.

- I consider the analyzes of figures 4 and 6 redundant. I believe one of them might go in the supplementary material.

- In figure 5, it would be interesting to have a marking of twice the median of nucleotide diversity. This marking serves to identify mutational hotspots, which, as it is, are being marked without a well-defined criterion.

- In figure 8 it would be interesting if the length of the branches appeared in the figure. Also, explain what the asterisks are (I understand what they are. the species you sequenced but it's not written).
